# The Potential of Edible and Medicinal Resource Polysaccharides for Prevention and Treatment of Neurodegenerative Diseases

**DOI:** 10.3390/biom13050873

**Published:** 2023-05-22

**Authors:** Qingxia Gan, Yugang Ding, Maoyao Peng, Linlin Chen, Jijing Dong, Jiaxi Hu, Yuntong Ma

**Affiliations:** 1College of Pharmacy, Chengdu University of Traditional Chinese Medicine, Chengdu 611137, China; 2State Key Laboratory of Traditional Chinese Medicine Processing Technology, State Administration of Traditional Chinese Medicine, No. 1166, Wenjiang District, Chengdu 611137, China

**Keywords:** neurodegenerative diseases, edible and medicinal resources, polysaccharides, health products, pathways

## Abstract

As natural medicines in complementary and alternative medicine, edible and medicinal resources are being gradually recognized throughout the world. According to statistics from the World Health Organization, about 80% of the worldwide population has used edible and medicinal resource products to prevent and treat diseases. Polysaccharides, one of the main effective components in edible and medicinal resources, are considered ideal regulators of various biological responses due to their high effectiveness and low toxicity, and they have a wide range of possible applications for the development of functional foods for the regulation of common, frequently occurring, chronic and severe diseases. Such applications include the development of polysaccharide products for the prevention and treatment of neurodegenerative diseases that are difficult to control by a single treatment, which is of great value to the aging population. Therefore, we evaluated the potential of polysaccharides to prevent neurodegeneration by their regulation of behavioral and major pathologies, including abnormal protein aggregation and neuronal damage caused by neuronal apoptosis, autophagy, oxidative damage, neuroinflammation, unbalanced neurotransmitters, and poor synaptic plasticity. This includes multi-target and multi-pathway regulation involving the mitochondrial pathway, MAPK pathway, NF-κB pathway, Nrf2 pathway, mTOR pathway, PI3K/AKT pathway, P53/P21 pathway, and BDNF/TrkB/CREB pathway. In this paper, research into edible and medicinal resource polysaccharides for neurodegenerative diseases was reviewed in order to provide a basis for the development and application of polysaccharide health products and promote the recognition of functional products of edible and medicinal resources.

## 1. Introduction

People cannot survive without food. It is clear that food security is a priority for many people. With the improvement in living standards, people have become more aware of their health. As people meet their nutritional requirements, they gradually turn their attention toward enjoying a green, safe, and healthy life. Edible and medicinal resources represent the combination of food and health, with food as a substitute for medicine. Edible and medicinal resources used in disease prevention are prioritized as nourishment first and treatment second [1,2]. Especially in our new situation characterized by a constantly changing epidemic, combined with the sub-optimal health situation that most people find themselves in, edible and medicinal resources highlight the significance of health maintenance, health care, and medical treatment. In recent years, edible and medicinal resources have been widely recognized and accepted by the international community for their use in the prevention and treatment of common, frequently occurring, chronic and serious diseases [3].

Neurodegenerative diseases such as Alzheimer’s disease (AD), Parkinson’s disease (PD) and Huntington’s disease (HD) are chronic diseases that are difficult to control with a single treatment [4]. With the rapidly increasing size of the aging population in our society, the incidence of neurodegenerative diseases, which are “chronic killers” in the elderly population, has risen sharply [5]. Patients who are supposed to enjoy a joyful old age after working hard throughout their lives have been deprived of memory, independence and dignity by neurodegenerative diseases that cause well-known pain and are an unbearable physical, emotional, and financial burden on patients and even more so on the whole family [6,7]. What is more frightening is that since brain cells cannot regenerate, the degeneration and death of neurons worsen over time, with devastating and irreversible consequences that ultimately end in death [8,9]. The World Health Organization has predicted that by 2050, neurodegenerative diseases will be the second-leading cause of death in humans [10]. Unfortunately, almost all neurodegenerative diseases today have no effective treatments that relieve symptoms [11]. Edible and medicinal health products inspire new ideas for the treatment of neurodegenerative diseases; their prevention and treatment with safe and effective health products seem to be an excellent approach. At present, many edible and medicinal resources, including *Panaxginseng*, *Lycium barbarum*, *Angelica sinensis*, *Codonopsis pilosula,* and so on, have been included in the diet of Chinese families for the maintenance of good health. Polysaccharides are some of the safest active ingredients in these edible and medicinal sources, and are the focus for developing safe, healthy, and green edible and medicinal health products [12]. Interestingly, the formation and aggregation of abnormal proteins such as Aβ, NFTs, α-synuclein, and mHtt that cause neuronal apoptosis and autophagy, oxidative damage, neuroinflammation, the release of neurotransmitters, and synaptic plasticity in neurodegenerative diseases were confirmed to be regulated by edible and medicinal polysaccharides (EMPs). The regulation of these abnormal proteins depends on the influence of the NF-κB, MAPK, Nrf2, mitochondrial, mTOR, PI3K/AKT, P53/P21, and BDNF/TrkB/CREB pathways. Therefore, we consider EMPs to be a promising candidate for the prevention and treatment of neurodegenerative diseases by targeting multiple signaling pathways. In this paper, the possible mechanisms and pathways of EMPs in the prevention of neurodegenerative diseases were reviewed to provide a basis for the development and application of polysaccharide health products and promote the recognition of functional products of medicinal and edible resources.

## 2. The Potential of EMPs for Regulating Neurodegenerative Diseases

Because of the complex and chronic nature of neurodegenerative diseases, multi-way long-term prevention and treatment with EMPs is effective. This is not only due to the improvement of the pathological features of the behavior, movement, and reduction of abnormal protein aggregation, but also due to the mechanism of regulating abnormal apoptosis and autophagy, alleviating oxidative damage, promoting the secretion of neurotransmitters, promoting synaptic plasticity, and reducing the inflammatory response.

### 2.1. Behavioural Movement Enhancement

Neurodegenerative diseases are usually characterized by motor or cognitive impairment [13]. AD symptoms include progressive memory and cognitive impairment, personality changes, and language impairment [14,15]. Resting tremor, bradykinesia, rigidity, and postural disturbances are common in PD [16,17]. HD is characterized by involuntary movements, mental disorders, and progressive dementia [18,19]. 

These behavioral changes have been partially demonstrated in animal models of neurodegenerative diseases. However, these changes were alleviated by EMPs. The impairment of short-term learning, memory, and cognition in AD mice was indicated in a series of behavioral experiments, including those where the researchers employed the Y-maze, Morris water maze, open field test, and novel object recognition test [20,21]. EMPs, such as *Inonotus obliquus* (IOP), *Amanita caesarea* (ACP), and *Ganoderma lucidum* (GLP), alleviated pathological behavior disorders according to the results of the aforementioned tests and reduced the effects of the spatial learning disabilities caused by AD [22,23,24]. For PD model mice, their coordination, cognitive, limb motor, and limb coordination ability were remarkably decreased compared with normal mice. The presence of complex polysaccharides mainly composed of galactose and rhamnose from *Momordica Charantia* considerably increased the behavioral test scores in rotation tests with longer dwell times, caused mice to take a shorter amount of time to reach the bottom during pole tests, and increased the motor ability of PD mice [25]. HD model TG mice exhibit phenotypes similar to those of HD patients, including a shortened lifespan, motor deficits, and weight loss. LBP, the main functional component of the thousand-year-old health food *L. barbarum*, extended lifespan, considerably reduced weight loss, and increased dwell time on the rotarod to promote the motor function of TG mice [26]. As shown in Appendix A, other EMPs were present, which also enhanced the behavior and motor function of mice, meaning that they show potential for anti-neurodegenerative diseases.

### 2.2. Reduction in Abnormal Protein Accumulation

The occurrence of neurodegenerative diseases is inseparable from the aggregation and deposition of misfolded proteins [27,28]. Proteins are one of the components of organisms, and they play an irreplaceable role in the normal growth and development of the body [29]. For neurons, which are a type of non-proliferating cells, the production of normal proteins is even more important to ensure their viability [30]. Once the protein forms the wrong structure, it will not only lose its biological function, but it will also trigger neuroinflammation, oxidative damage, neuronal apoptosis, and autophagy and will eventually lead to the loss of neurons [31,32]. Whether the cause is genetically familial or multifactorial late-onset AD, the main neuropathological features are the extensive extracellular deposition of insoluble amyloid Aβ formed by the hydrolysis of APP by β and γ hydrolases and the intracellular neurofibrillary tangles (NFTs) formed by hyperphosphorylated tau protein [33,34]. Similarly, the major neuropathological features of PD are inseparable from intracellular Lewy bodies composed of aggregates of the misfolded presynaptic protein α-synuclein (α-syn) [35,36,37]. HD is caused by a mutated gene that produces mHTT, which is a stretch of a protein called polyQ that is incorrectly stretched and alters the natural form and function of HTT [38,39].

In AD mice, excessive Aβ deposition in the hippocampus was remarkably reduced after chronic polysaccharides were administrated, as shown in Appendix A. Abnormal Aβ production first depends on the cleavage of APP by β-secretase. The soluble polysaccharides (CPP) of *Codonopsis codonopsis*, which are commonly found in Chinese diets, reduced BACE1 (β-secretase) activity and inhibited Aβ_1–42_ production to reduce deposition in the hippocampus of APP/PS1 mice and N2a-APP cells [40]. In addition to Aβ_1–42_, the aggregation of NFTs caused by high levels of p-Tau was observed in the brains of AD mice and rats. Fortunately, the aggregates were strongly reduced after treatment with EMPs [22,41,42]. Related mechanisms suggested that the attenuation of tau hyperphosphorylation by CPP was strongly associated with PP2A activation, which is the major dephosphorylase of tau [43]. In addition, EMPs are also effective against PD and HD. The level of aggregated α-syn is considerably increased in PD mice compared with normal mice. LBP treatment effectively suppressed this aggregation, confirming the effectiveness of anti-PD [44]. LBP also remarkably reduced mHtts in the cortex, hippocampus, and striatum, which showed the potential to ameliorate HD [45]. In addition, the accumulation of polyQ in the AM141 *Caenorhabditis elegans* model was inhibited by polysaccharides from medicinal and edible *Astragalus membranaceus*, thereby reducing neurotoxicity [46].

### 2.3. Neuronal Apoptosis Inhibition

The massive loss of functional neurons is a major cause of neurodegenerative diseases. Abnormal apoptosis affects neuron loss. In AD, DNA fragmentation and pro-apoptotic protein upregulation, mitochondria damage, and increased caspase-3 expression in the hippocampal region result in a loss of cholinergic neurons [47,48]. The pathological symbol of PD is the loss of dopaminergic neurons primarily caused by apoptosis [49]. The massive α-syn aggregation is critical for the ability to promote neuronal death as it affects the mitochondrial respiratory chain and activates caspases-9 and -3 [50]. In addition, mutations in many genes associated with familial PD, such as parkin and PINK1, suppress protective properties and increase the sensitivity of dopaminergic neurons to apoptosis [51]. Protein mHTT in HD results in the decreased expression of respiratory chain enzymes and a loss of MMP. The subsequent cytochrome C release and caspase activation promote the cleavage and translocation of mHTT into the nucleus to aberrantly interact with multiple transcription factors, including p53, to determine further striatal neuronal apoptosis [52].

EMPs inhibited apoptosis, thereby reducing loss in relevant neurons in neurodegenerative diseases. Morphological changes in apoptosis induced by H_2_O_2_ were observed under DNA dye Hoechst 33342 staining, including the neurodegeneration and shrinkage of cell bodies, as well as the fragmentation and condensation of the nucleus, which occurred in 22.9% of the total cells. Sulfated hetero-polysaccharides (UF) from *Saccharina japonica*—a seafood used in traditional Chinese medicine Kunbu—reduced the number of apoptotic and dead cells; especially, 500 g/mL of UF greatly reduced apoptosis to 10.7% [53]. In addition, the anti-apoptotic properties of the EMPs were demonstrated by experiments in vivo. Treatment with *A. sinensis* polysaccharides (ASP), which are found in medicinal diets, considerably reduced the increase in the number of TUNEL-positive neurons in AD rats [45]. After treatment with polysaccharides (PSK) from *Trametes versicolor*, one of the most medicinal fungi, NeuN (neurons) and MAP2 (dendritic) staining areas were significantly increased, and neuronal apoptosis, which was detected by caspase-3 in the hippocampus, was reduced in APP/PS1 mice [54]. Ensuring the integrity of the mitochondria has always been the key to preventing apoptosis. In the brain tissue of PD mice, considerably increased levels of cytochrome C and mitochondria-related apoptotic factors such as Bax were observed; however, treatment using *M. Charantia* polysaccharide (MCP) treatment remarkably reversed the expression changes of these proteins, thereby suggesting anti-apoptotic effects [25].

### 2.4. Reduction in Neuronal Oxidative Stress

Under normal conditions, most of the oxygen absorbed by the body is reduced to water and is converted into energy through mitochondrial respiration [55,56]. However, a small amount (≤2%) of oxygen can accept one or two electrons in the middle of the respiratory chain and can be partially reduced to generate ROS [57]. To maintain physiological balance, excess ROS is removed through enzymatic and non-enzymatic defense systems at any time [58]. However, when the physiological environment changes, such as by aging, hypoxia, or other diseases that damage the mitochondrial respiratory chain, the generation and accumulation of ROS are further increased, or the activity of antioxidant enzymes is affected so that the imbalance between the oxidative system and the antioxidant system cause oxidative damage [59,60]. Neurodegenerative diseases are at least associated with aging, and oxidative stress is naturally inevitable [61]. In AD, accumulated ROS promotes the cleavage of APP to Aβ by enhancing γ- and β-secretase activity, which results in an increased Aβ deposition [55]. Aβ, in turn, induces oxidative stress from multiple pathways, such as by interacting with catalase (CAT) in the brain to impair the ability to scavenge ROS. This vicious cycle eventually leads to continuous damage to neurons [62]. In the brains of PD patients, genetic abnormalities or environmental toxins lead to increased concentrations of free dopamine that are auto-oxidized, and the product eventually splits into a large amount of OH- via catalysis, which results in the exposure of dopaminergic neurons to oxidation under stress [63]. Furthermore, free dopamine inhibits the transition of α-syn from fibrils to mature fibrils, which leads to the accumulation of soluble fibrils in dopaminergic neurons, which is a hallmark of PD [64]. This horrific vicious cycle occurs in PD patients, whereby the accumulation of α-syn leads to a decrease in the number of vesicles, which in turn increases free dopamine and enhances oxidative stress [65]. Relatively few studies have been conducted on HD, but we know that it is correlated with oxidative stress. The levels of oxidative damage products were considerably elevated in the corresponding degenerated regions in the brains of HD patients. Aggregated HTT directly leads to increased ROS production, which causes mHTT-expressing cells to die before normal cells [66].

ROS levels in neurons were remarkably increased under various inductions, such as Aβ_1–40_, L-Glu, paraquat, rotenone, and gene mutation, and EMPs reversed the excessive ROS accumulation and reduced the oxidative damage. In addition to the direct inhibition of ROS production in response to fungal *Fomes officinalis* Ames polysaccharides (FOAP) against oxidative stress, the subsequent elevated SOD activity and reduced MDA levels in Aβ_25–35_-treated PC12 cells were also key pieces of evidence of the effectiveness of EMPs [67]. Additionally, by successfully increasing the SOD activity and decreasing the MDA content, paraquat-exposed *C. elegans* survived considerably longer after treatment with *Epimedium brevicornum* polysaccharides [68]. Moreover, low SOD and GSH-Px levels and very high ROS and MDA levels were observed in the serum and whole brain of APP/PS1 mice, and these phenomena were strongly reversed by 8 weeks of IOP administration [22]. In addition, mitochondrial-basal respiration and respiratory chain complex 1 were inhibited in the midbrain tissue of rotenone-induced PD rats. However, fucoidan ameliorated the above symptoms to reduce the possibility of ROS production and modulate the increase in the content of three oxidative stress products: MDA, 3-NT, and 8-OhdG [69].

### 2.5. Neuroinflammation Inhibition

The neuronal damage and loss in neurodegenerative diseases are inextricably linked to the chronic activation of the innate immune response in the central nervous system [70,71]. The immune cells in the brain are mainly microglia, which are innate immune cells in the brain parenchyma that can respond to traumatic injury or inflammatory signals to protect the brain and act as sensors for various environmental signals [72,73]. However, the persistence of the activation signal or the failure of the repair mechanism can lead to the continuous activation of microglia, which results in the release of excessive cytotoxic factors, which then results in prolonged and persistent neuronal death [74,75]. The massive deposition of abnormal proteins such as Aβ, NFTs, α-syn, and mHTT is not efficiently cleared by microglia, but instead continuously triggers the secretion of pro-inflammatory cytokines from these cells. These pro-inflammatory cytokines subsequently lead to the loss of multiple neurons and also promote the accumulation of abnormal proteins, which aggravate neurodegenerative diseases [76,77,78].

The anti-inflammatory activity of EMPs was first demonstrated by reversing a remarkable increase in the number of astrocytes and microglia in neurodegenerative mice [54,79]. The polysaccharides from ginseng, a rare medicinal herb, regulate immunity, and related healthcare products containing these polysaccharides have been developed. The active polysaccharide NFP of Korean red ginseng remarkably inhibited the increase in the IBA-1(+) area of AD mice, which is a marker of microglia cells [80]. Moreover, the expression of Emr1 associated with microglial activation in MPTP-induced PD mice was inhibited after treatment with algal polysaccharides from *Chlorella pyrenoidosa* (CPS) treatment [81]. A remarkable elevation in the inflammatory factors IL-1β, IL-6, and TNF-α was observed in AD and PD mice. In contrast, most EMP treatments inhibited the levels of these factors [25,54,79,82]. The polysaccharide ATP from *Acorus tatarinowii*, which is used to treat forgetfulness and insomnia, inhibited the production of inflammatory cytokines in LPS-stimulated BV2 microglia and thus counteracted the effect of LPS on nitrite production, meaning it exhibited anti-neurodegenerative potential. The ATP-induced reduction in iNOS transcriptional expression considerably suppressed the inflammatory mediator NO production and pro-inflammatory COX-2 mRNA expression [83]. It has been reported that 6-OHDA further activated the expression of inflammasome NLRP3 by inducing ROS. However, the intervention of polysaccharides (ACP) from precious medicinal edible *Antrodia camphorata* suppressed NLRP3 expression to reduce toxicity in MES23.5 cells and PD mouse-like lesions [84,85]. The disruption of intestinal barrier function, which is often caused by gut microbe disorder, promotes the entry of bacterial-derived pathogens and endotoxin LPS into the circulatory system, leading to neuroinflammation and even neurological disorders [10]. The β-glucan, SCP-1 from health food of *Sparassis crispa* inhibited the activation of microglia and astrocytes in the brains of AD-like mice and down-regulated IL-6, IL-1β, and TNF-α levels, which may be attributed to the fact that SCP-1 significantly increased the expression of ZO-1 and occludin in the colon and decreased LPS levels [86]. Similarly, the polysaccharides (PSP) of the supplement *Polygonatum sibiricum* regulated AD symptoms by decreasing the inflammatory environment in the same way [87].

### 2.6. Gut Microbiota Regulation

Neurodegenerative disease is associated with early gastrointestinal motility abnormalities, including constipation and delayed gastric emptying, and patients also have an imbalance of intestinal microbiota [88]. Subsequently, the interrelationship between gut microbiota and neurodegenerative diseases has also been studied and demonstrated in various animal models [89,90,91]. Disturbances in the gut microbiota at least exacerbate the pathological development of neurodegenerative diseases [92]. This is because the gut microbiota affects the secretion of metabolites, such as short-chain fatty acids, which are related to the regulation of nervous immune function, sensory nerve signals, and metabolic activities of the central nervous system [93,94,95]. In addition, disturbances in the gut microbiota often impair the intestinal barrier, elevate lipopolysaccharide, and increase abnormal proteins, leading to neuroinflammation in neurodegenerative diseases [96,97]. Therefore, pathological changes in PD, AD, and HD patients may be improved by regulating the gut microbiota with drugs.

Gut microbiota have a clear preference for polysaccharides, which may be beneficial for the growth of specific species and may play a role after fermentation by microbiota. Changes in the gut microbiota of AD mice at the phylum level that were induced by AlCl_3_ and D-Gal were characterized by lower levels of Firmicutes and an increased abundance of Bacteroidetes, which may be related to Aβ accumulation in the brain and systemic inflammatory status in patients with cognitive deficits. SCP-1 reversed this phenomenon by reshaping the composition of the gut microbiota. Moreover, SCP-1 also significantly promoted the growth of *Intestinaimonas*, *[Eubacterium] ventriosum group*, *Lachnospiraceae_UCG_010,* and *Lachnospiraceae_UCG_001*, and promoted the synthesis of short-chain fatty acids to maintain intestinal integrity, improve cognitive function, and regulate the immune response [86]. The study investigated that the abundances of the three major inflammation-stimulating bacteria *Helicobacter typhlonius*, *Helicobacter mastomyrinus*, and *Akkermansia muciniphila* decreased significantly after PSP was orally administered to 5 × FAD mice for 3 months, thus alleviating neuroinflammatory stress. Furthermore, PSP significantly inhibited the reduction in *muciniphila,* which was negatively associated with neurodegenerative diseases [87].

### 2.7. Autophagy Regulation

Autophagy plays a dual role in neurodegenerative diseases [98]. Studies have shown that dysfunction of autophagy led to disturbance of liposome-mediated degradation pathways, and thereby blocked the ability of lysosome to eliminate macromolecules and damaged organelles, which resulted in the release of cytochrome c and other pro-apoptotic proteins [99]. The expression levels of proteins LC3-II and Beclin often reflect the level of autophagy. In 6-HODA-induced PC12 cells, the expression of LC3-II was decreased, which suggested the impairment of autophagy, which was reversed by APS treatments [100]. Conversely, the overactivation of autophagy increased dopaminergic neuron degeneration. The protein levels of LC3-II and Beclin were abundantly expressed in MPTP-induced PD mice and were significantly downregulated by LBP treatment to suppress the degeneration of neurons in SN [44].

### 2.8. Regulation of the Balance of Neurotransmitters

The balance of neurotransmitters reduces the risk of developing neurodegenerative diseases [101]. An imbalance of neurotransmitters has been found in AD patients; for example, a reduction in inhibitory amino acids (GABA and Ach) and an increase in the excitatory amino acid (Glu) is strongly associated with dementia and cognitive impairment. Fortunately, the concentrations of these neuroactive substances tended to return to normal levels after high-dose intervention of SCP [42]. Simultaneously, DA is the neurotransmitter of most concern in PD. In MPTP-induced PD mice, the immunoreaction and mRNA expression of DA-initiating and rate-limiting enzymes TH and DAT were decreased, which suggests a major loss of DA. Polysaccharides from *Spirulina platensis* (PSP1) pretreatment significantly increased DA levels and decreased dopamine metabolic rate [102]. Similarly, CPS protected the depletion of striatal DA and TH-positive neurons in SN and increased the transportation of DA, significantly attenuating neurotoxin-induced behavioral impairment [81]. Furthermore, elevated levels of DA and 5-HT were ameliorated after MCP treatment in response to brain dysfunction under PD pathological conditions [25].

### 2.9. Restoration of Synaptic Plasticity

The loss of synapses and the reduction in synaptic plasticity in hippocampal neurons may be one of the pathological features of AD and the neurobiological basis of learning and memory dysfunction [103]. It is reported that LBP promoted neurogenesis and restored hippocampal synaptic plasticity in APP/PS1 mice [104]. Moreover, CPP enhanced APP/PS1 synaptic plasticity, repaired synapses, and reduced cognitive deficits, and these effects were possibly associated with a significantly increased expression of synaptic proteins PSD95 and synaptotagmin [40]. DCX is a hallmark of adult neurogenesis, and NFP treatment significantly affected the number and dendritic morphology of DCX(+) neurons in healthy and AD brains, stimulated neurogenesis, and increased connectivity between hippocampal neurons [80].

## 3. Multiple Pathways Support the Regulation of Neurodegenerative Diseases with MEPs

The abovementioned ability of MEPs is mainly regulated through multiple targets and pathways. The mitochondrial, NF-κB, MAPK, Nrf2, mTOR, PI3K/AKT, P53/P21, and BDNF/TrkB/CREB pathways can be regulated by MEPs to prevent and treat neurodegenerative diseases. The main related mechanisms and pathways of EMPs against neurodegenerative diseases are shown in Figure 1, Figure 2 and Figure 3.

### 3.1. Mitochondrial Pathways

The mitochondrial pathways, which are apoptotic pathways triggered by a variety of stress conditions and drugs, are severe pathways for functional neuronal loss in neurodegenerative diseases [105]. An abnormal respiratory chain reaction caused by a variety of abnormalities leads to the enhancement of MMP, which causes the release of cytochrome C to further activate the initial caspase and then activate terminal caspase to induce cell apoptosis. When the apoptotic signal is transduced to the mitochondria, the pro-apoptotic proteins Bad, Bid, Bax, and Bim of the Bcl-2 family are transferred from the cytoplasm to the mitochondria and are combined with the anti-apoptotic protein Bcl-xL, thereby increasing the MMP. Other signals, such as ROS, directly trigger the opening of mitochondrial inner membrane pores, leading to the rupture of the outer mitochondrial membrane and the outflow of cytochrome C [106]. A variety of polysaccharides from medicinal and edible fungi such as edible *Dictyophora echinovolvata* (DEVP), *Tremella fuciformis*(TF04), *Armillaria mellea* (AMPS), *Fomes officinalis* Ames (FOAP), and *S. crispa* (SCWEA) were found to inhibit the mitochondrial apoptotic pathway against neurodegenerative diseases. H_2_O_2_ is commonly used to induce a considerable increase in the Bax/Bcl-2 ratio, cytosolic cytochrome C, and cleaved caspases-3 levels in PC12 cells. The inhibitory effect of DEVP is achieved, at least in part, by inhibiting the mitochondrial apoptotic pathway [107]. Exposure to glutamate strongly increases Bax expression, cytochrome C release, and the activities of caspase-8, -9, and -3; however, a remarkable reversal was observed after TL04 pretreatment, which inactivated the caspase-dependent mitochondrial pathway to alleviate damage to PC12 cells [108]. SCWEA, in L-glu-induced PC12 cells, restored the normalization of the expression of the anti-apoptotic protein Bcl-2 and Bcl-xL, which indicated a protective effect on neurons [109]. The autonomic activity of D-gal and AlCl_3_-induced AD mice was enhanced after 4 weeks of AMPS administration, which was inseparable from the inhibition of mitochondrial-mediated apoptosis [110]. FOAP potently inhibited Aβ_25–35_-induced cytotoxic effects, thereby attenuating apoptosis, increasing the ratio of Bcl-2/Bax, and inhibiting the release of cytochrome C from mitochondria to the cytoplasm in PC12 cells. FOAP remarkably alleviated mitochondrial dysfunction by regulating MMP and promoting the synthesis of mitochondrial ATP [67]. In addition, polysaccharides from *Gynostaphyllum pentaphyllum*, *Taxus chinensis,* and other medicine on the list of health products also save mitochondrial pathways, as shown in Appendix A.

### 3.2. MAPK Pathway

The MAPK family consists of serine-threonine protein kinases that are widely distributed in mammalian cells, whose members play key roles in neuronal inflammation, proliferation, differentiation, survival, and death [111]. For example, ERK, JNK, and p38 pathways are involved in regulating the synthesis and release of pro-inflammatory cytokines in microglia; the activation of the ERK pathway is necessary for neuronal proliferation, survival, and differentiation, and the activation of JNK is thought to regulate neuronal death, especially apoptosis. The MAPK pathway can be regulated by EMPs. Fucoidan could inhibit LPS-activated microglia, which is manifested in the suppression of the production of NO, the expression of iNOS, and the morphological transformations by inhibiting the expression of p38 and ERK pathway-related proteins [112]. Polysaccharides from *Morchella importuna* inhibited H_2_O_2_-induced PC12 cell apoptosis by downregulating the p38-JNK pathway, as well as activating the ERK to enhance Bcl-2 expression, reduce Bax expression, and decline caspase-3 [113]. In addition, in Aβ_1–42_ peptide-induced AD mice, SCP considerably improved the changes in their cognition and histopathology, the deposition of Aβ, the expression of pro-inflammatory cytokines, and the activation of astrocytes and microglia, which were also associated with the MAPK pathway, which is involved in inhibiting the phosphorylation of p38, JNK, and ERK [79].

### 3.3. NF-κB Pathway

The transcriptional factor NF-κB regulates the expression of a series of inflammatory genes and plays an important role in various cellular inflammatory responses. In microglia, when stimulated by inflammatory signals, such as LPS, the stable NF-κB with its inhibitor IκB-𝛼 complex in the cytoplasm will liberate NF-κB to the nucleus, regulate the expression of TNF-α and other genes and cause neuronal inflammatory responses [114]. EMPs have been found to block the activation of NF-κB, which showed their potential to improve several neurodegenerative diseases. APS, known as an immunostimulant, suppressed the NF-κB and AKT signaling pathway to reduce LPS-stimulated NO, PGE2, the generation of the pro-inflammatory cytokines IL-1β and TNF-α generation, and iNOS and COX-2 gene expression [83]. The TLR4/MyD88 and PI3K/AKT pathway are the main upstream of the NF-κB pathway, and ATP significantly reduced the abnormal rise in inflammatory cytokines in LPS-induced BV2 cells by reversing the up-regulation of proteins, which provided neuroprotection against inflammation-induced neurotoxicity [115]. Other inducers, such as Aβ_1–42_ peptide-induced AD mice, also showed increased NF-κB and decreased IκB-α, whereas the administration of polysaccharides SCP from *Schisandra Chinensis* decreased the nuclear translocation of NF-κB, thereby reducing the expression and release of pro-inflammatory cytokines [79]. MPTP treatment significantly promoted the expression of TLR4, MyD88, and p-p65 proteins, while MCP inactivated the TLR4/MyD88/NF-κB pathway and exerted anti-inflammatory effects in PD [25].

### 3.4. Nrf2 Pathway

The transcription factor Nrf2 regulates a series of antioxidant enzymes involved in oxidative stress-related neuronal dysfunction, thereby aggravating the pathogenesis of neurodegenerative diseases [116]. Under physiological conditions, the combination of Nrf2 and keap1 is inactivated in the cytoplasm. After being stimulated and activated, Nrf2 is liberated from keap1 and is subsequently translocated to the nucleus, which leads to the activation of a series of antioxidant enzymes to exert antioxidant effects. Several EMPs have been found to activate Nrf2 to reduce neuronal oxidative damage. For instance, IOP enhanced the expression levels of Nrf2 and its downstream proteins, including HO-1 and SOD-1, in L-Glu-induced HT22 cells and the brains of APP/PS1 mice [22]. APS up-regulated the expression of Nrf2 in the nucleus in brain tissues of APP/PS1 mice and restored the expression levels of antioxidant enzymes SOD and GSH-Px [117]. In vivo and in vitro experiments showed that the reduction in Nrf2 and NQO1, which are anti-oxidative stress-related proteins, although induced by MPTP, was restored after PSP administration, which indicated the anti-dopaminergic neurodegeneration ability of PSP [118]. Rotenone decreased the expression of the PGC-1α and Nrf2 proteins in the ventral midbrain, whereas their expression was significantly up-regulated by Fucoidan, which may explain the protective effect of Fucoidan on mitochondrial function [69].

### 3.5. mTOR Pathway

The mTOR is a central cell growth regulator whose phosphorylation and dephosphorylation lead to the inhibition and induction of autophagic death under different conditions [119]. mTOR is often regulated by its upstream pathway AKT, whereas the inactivation of the AKT/mTOR signaling pathway impairs neuronal function and leads to neuronal autophagic death after injury. PSP had the ability to prevent MPP+-induced death of neuronal injury in vivo and in vitro by activating the AKT/mTOR pathway [118]. Autophagy, as a survival-promoting self-defense strategy, also plays an important role in reducing oxidative stress. Polysaccharides from *Hericium erinaceus* positively regulated mTOR, and this regulation was dependent on AKT activity, thereby inhibiting CaMK II/IV phosphorylation-related oxidative stress-mediated calcium homeostasis, ultimately improving AD symptoms [41]. In addition, PTEN is a negative regulator of the mTOR pathway that reduced the activation of AKT and prevented all downstream signal transduction events regulated by AKT. LBP treatment up-regulated the phosphorylation of AKT and mTOR and may exert an anti-autophagic effect by activating the PTEN/mTOR signaling axis in the SN to alleviate the excessive autophagy and loss of dopaminergic neurons in the SN in MPTP-induced mice [44]. The hyperactivation of the mTOR signaling pathway leads to severely impaired autophagy, which results in the accumulation of abnormal proteins in neurons, which is a major feature of neurodegenerative diseases. This phenomenon was observed in 6-HODA-induced PC12 cells, and it was gratifying that APS treatment inhibited the AKT/mTOR signaling pathway to promote the conversion of LC3-I to LC3-II, improve the formation of autophagosome and increase cell viability and the level of autophagy [100].

### 3.6. PI3K/AKT Pathway

AKT, as a proto-oncogene, has become a hot spot of interest due to its ability to regulate various downstream targets for various neuronal functions, including regulation of inflammation, oxidative stress, apoptosis, and autophagy [120]. Recent studies have shown that LBP up-regulated miR-4295 in H_2_O_2_-injured HTM cells to activate the PI3K/AKT signaling pathways, which are involved in regulating oxidative damage in HTM cells [121]. GSK-3β is the downstream pathway of PI3K/AKT, which is involved in and affects the regulation of tau protein synthesis and dopamine signaling, and directly triggers apoptosis signals and other downstream events. By initiating the Shh and PI3K/AKT signaling pathway, increasing p-GSK-3β, and inhibiting GSK-3β activity, TMT-induced neurotoxicity in N2a cells is antagonized by LBP [122]. Interestingly, anti-HD potential was demonstrated in LBP as a result of reducing mHtt in the cortex, hippocampus, and striatum of TG mice again by activating AKT [123]. In addition, OP treatment inhibited the decreased expression levels of PI3K, p-PI3K, AKT, and p-AKT in the hippocampus of Aβ_1–42_-induced AD model mice and increased the expression level of p-GSK-3β to improve metabolic function and cognitive impairment [20]. L-Glu-induced AKT inhibition and subsequent GSK-3β phosphorylation were also associated with the promotion of mitochondria-related pro-apoptotic stimuli. These phenomena recovered after SCWEA administration [109]. The study revealed that GLP potentiated activation of FGFR1 and downstream ERK and AKT cascades, promoting neurogenesis upon growth factor deficiency, and had the potential to serve as a preventive and therapeutic agent against neurodegenerative diseases [24]. Aβ_25–35_ treatment remarkably reduced the protein expression of p-AKT in PC12 cells, whereas the pretreatment with PSP revealed its anti-apoptotic properties by enhancing the PI3K/AKT pro-survival pathway to play a neuroprotective effect [124]. Conversely, EMPs, such as APS and ATP, had inhibitory effects on the AKT pathway because the hyperphosphorylation of AKT promoted the downstream NF-κB pathway and mTOR pathway, stimulating inflammation and autophagy disorders [100,115].

### 3.7. P53/p21 Pathway

The P53 protein induces cell cycle termination or apoptosis, which is known as cell senescence [125]. The level of P53 protein is low in normal cells, but in neurodegenerative diseases, the post-transcriptional modification pathway of p53 is directly activated, and the expression of P53 protein in the nucleus is increased. At this time, p53 can activate the downstream signaling molecule p21, which plays a role in cell cycle arrest, differentiation, and apoptosis [126]. The expression of protein P53 and P21 was up-regulated in D-gal-induced AD mice; however, the ASP-induced inactivation of p53 and the target genes p21 prolonged the lifespan of the mice and reduced the oxidative damage and inflammation in AD mice and thus resisting neurodegenerative diseases [127].

### 3.8. BDNF/TrkB/CREB Pathway

BDNF is the most widely distributed and most abundant neurotrophic factor in the mammalian brain, and it is widely expressed in the central nervous system [128]. BDNF often combines with Tyrosine Kinase B (TrkB) with a high affinity to exert biological effects [129]. TrkB, as a specific receptor of BDNF, activates the MEK/ERK/RSK and PI3K/Akt signaling pathways and promotes CREB phosphorylation, thereby activating genes related to long-term memory, promoting the expression of synapse protein and synaptic vesicles proteins, inducing the enhancement of long-term memory, and affecting the memory function of the brain by regulating synapse protein synthesis, changing the morphology of dendrites and spines, and enhancing synaptic activity, which is of great significance for the development of the nervous system [130,131,132]. It was found that the expression of BDNF, TrkB, p-Akt, and p-CREB in AD rats induced by Aβ_25–35_ was decreased, and this was ameliorated by ASP treatment. ASP affects the learning and memory processes of AD rats by activating the BDNF/TrkB/CREB pathway, which involves the main transcription factors for brain development, neural survival, and neurogenesis [45].

## 4. Conclusions and Prospects

Polysaccharides are one of the most important active ingredients in medicine and edible resources for the maintenance of human health. Some EMPs have been used in health products on the market, such as *Panaxginseng*, *A. membranaceus*, and mushroom polysaccharides, in the form of oral liquids, capsules, and drinks. However, other active polysaccharide health products have seldom been developed. Additionally, polysaccharide health products are not unique or highly functional. The development premise of polysaccharide health products mainly involves supplementing deficiency and restoring normal conditions, without mentioning which organs or diseases cause deficiency and without targeting special consumer groups. In the age of aging, brain deficiencies caused by neurodegenerative diseases affect the health of an increasing number of people. Polysaccharides from edible and medicinal sources target neurodegenerative diseases and have the potential to develop related functional products. Therefore, scientific evidence of the anti-neurodegenerative effects of these EMPs was reviewed, and this will lay the foundation for the development of relevant functional health products for the long-term control and prevention of neurodegenerative diseases. Under the intervention of EMPs, the main manifestations of neurodegenerative diseases, including memory and cognitive impairment, motor retardation, and pathological features, such as abnormal protein Aβ, NFTs, α-syn, and mHtt accumulation, were improved, depending on at least one of the following aspects: (1) By promoting AKT phosphorylation, inhibiting the MAPK pathway, inhibiting the expression of pro-apoptotic Bcl-2 family proteins and caspase3/9, and repairing mitochondrial defects, the EMPs reduced the abnormal apoptosis of functional neurons. (2) By bidirectionally regulating mTOR phosphorylation expression, the EMPs not only reduced the excessive autophagic loss of neurons, but also prevented the accumulation of abnormal proteins in the neurons caused by impaired autophagy. (3) The EMPs alleviated ROS production and accumulation and promoted the expression of antioxidant enzymes, such as SOD, GSH-Px, HO-1, and GCLC, via activating the Nrf2 pathway, inhibiting neuronal oxidative damage. (4) The EMPs inhibited the expression of hyperphosphorylated AKT and prevented TLR4 activation, thereby inactivating NF-κB and MAPK pathways and inhibiting the release of TNF-α, IL-1β, IL-6, NO, iNOS, and other inflammatory cytokines in activated microglia. (5) EMPs regulated the release of neurotransmitters ACh, DA, DOPAC, HVA, 5-HT, and 5-HIAA and increased the activity of synaptic proteins to improve synaptic plasticity, exhibiting a protective effect on the nervous system.

However, the development of polysaccharide products still faces many challenges. First, the structure is the foundation. In nature, polysaccharides are mixtures with complex structure and variety. Determining the separation, purification, and structure of polysaccharides is difficult, and different extraction methods will have a certain influence on the structure of polysaccharides. In order to provide solid scientific and theoretical guidance for the extensive processing and development of functional products, further studies of the precise high-order structure and structure–activity relationships of MEPs, as well as the precise molecular mechanisms of their biological activity, are still necessary. Second, the effect is the premise. Many animal and clinical trials are necessary. Due to the difficulty in modeling brain diseases, current researchers mainly focus on in vitro mechanism studies and animal and clinical trials have seldom been conducted to determine the effect of polysaccharides on neurodegenerative diseases. Third, the clear mechanism of action is the direction. As macromolecular components, the way that polysaccharides function in the body has always been focused on. At present, researchers explored polysaccharides mainly focus on the structure and activity of polysaccharides and less research is conducted on the digestion, absorption, and glycolysis of polysaccharides in vivo. Because most polysaccharide products are still orally administered, it is not clear whether the structure and corresponding activity will be changed due to degradation by various enzymes and acidic conditions in vivo and whether the use of digested polysaccharides plays a role in absorption or intestinal microbial glycolysis. For functional products, the way the drug works is the key to determining the type of products, whether absorbable or prebiotic.

## Figures and Tables

**Figure 1 biomolecules-13-00873-f001:**
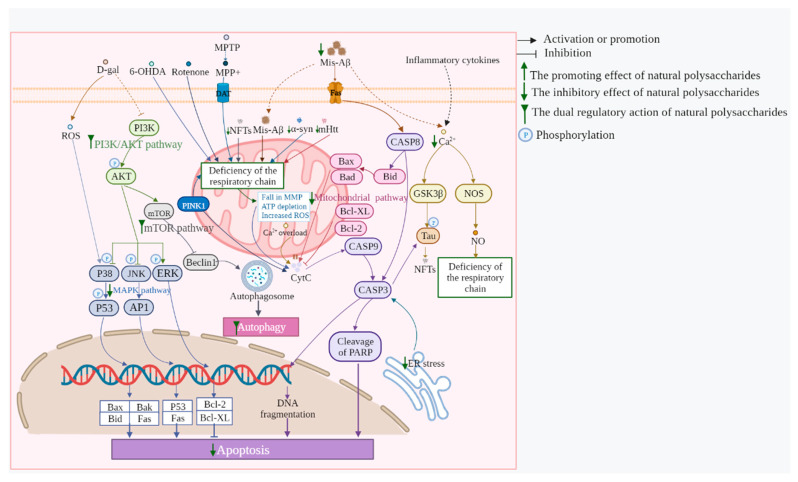
Anti-apoptosis mechanisms of polysaccharides in regulating neurodegenerative diseases.

**Figure 2 biomolecules-13-00873-f002:**
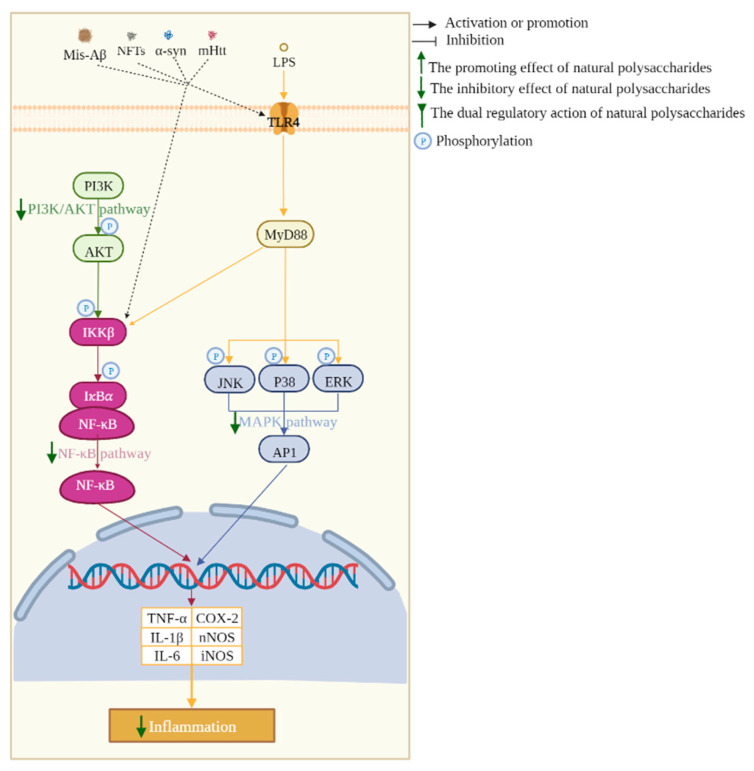
Anti-inflammation mechanisms of polysaccharides in regulating neurodegenerative diseases.

**Figure 3 biomolecules-13-00873-f003:**
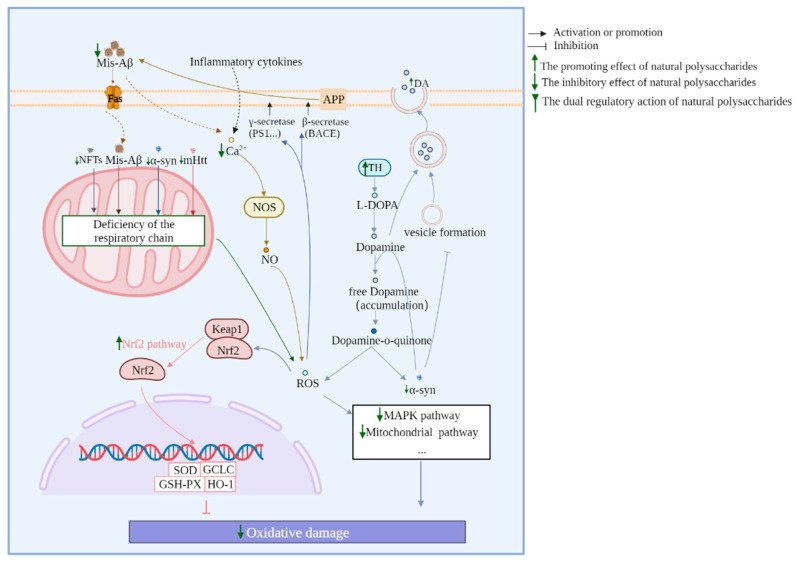
Anti-oxidation mechanisms of polysaccharides in regulating neurodegenerative diseases.

## Data Availability

Data sharing not applicable.

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
