# Peer review of "The Potential of Edible and Medicinal Resource Polysaccharides for Prevention and Treatment of Neurodegenerative Diseases"

_biomolecules, 2023, doi:10.3390/biom13050873_

Round 1
Reviewer 1 Report
ACCEOTED AS IT IS.
Author Response
Dear reviewer,
Thank you for your approval of this article.
Reviewer 2 Report
In my opinion, the article should not be accepted in its current form. The topic to which it relates is indeed interesting, and a review article in this area is well justified. However, in my opinion, the submitted work contains too many errors. The manuscript presents more of the processes or mechanisms that are involved in the development of neurodegenerative diseases, and there is little information about the effect of polysaccharides on these processes. I do not know on what basis the polysaccharides presented in the paper were selected. There is no information about their chemical structure. The way the results of the studies on polysaccharides are presented is not very clear. A better solution would be a summary in a table for specific diseases, for example. References to literature presenting the results of previous studies are missing in several places. There are errors in the Latin nomenclature of species in the paper. Krestin is not the name of a species of fungus but a specific polysaccharide extracted from Trametes versicolor. The language in the paper definitely needs to be corrected and improved, as many sentences are difficult to understand.
Author Response
Dear reviewer,
I am very grateful to your comments for the manuscript. According with your advice, we amended the relevant part in manuscript. We sincerely hope that the revision meets your satisfaction. Here below are our responses to your comments.
1)The manuscript presents more of the processes or mechanisms that are involved in the development of neurodegenerative diseases, and there is little information about the effect of polysaccharides on these processes. I do not know on what basis the polysaccharides presented in the paper were selected. There is no information about their chemical structure. The way the results of the studies on polysaccharides are presented is not very clear. A better solution would be a summary in a table for specific diseases, for example.
Response: Thank you for your comments. Per your suggestion, we have increased the information on polysaccharides and reduced the mechanism of neurodegenerative diseases appropriately in pages 3-7 of the article. We summarized the sources, structural information, models, administration methods, main therapeutic targets and pathways of polysaccharides for neurodegenerative diseases, and arranged them in the table, see Table 1 in the Supplementary. If we still lack information, we will add again.
The revisions were highlighted with red.
2)References to literature presenting the results of previous studies are missing in several places.
Response: We were really sorry for our mistakes and thank you for your reminder. We have reviewed the full text and added references where they are missing in pages 8-15 of the article.
The revisions were highlighted with red.
3)There are errors in the Latin nomenclature of species in the paper. Krestin is not the name of a species of fungus but a specific polysaccharide extracted from Trametes versicolor.
Response: We were really sorry for our mistakes and thank you for your reminder. We corrected the Latin for Krestin as Trametes versicolor in page 5 of the article, and checked all the other Latin.
The revisions were highlighted with red.
4)The language in the paper definitely needs to be corrected and improved, as many sentences are difficult to understand.
Response: We were really sorry for our mistakes. We have invited English majors to make revisions and have tried our best to polish the language in the revised manuscript. If there are still many errors, we can send it to an agency that specializes in English language editing and writing of scientific papers to assist us to re-write it properly.
The revisions were highlighted with red.
Reviewer 3 Report
The review addresses a compilation of evidence against neurodegenerative diseases from the polysaccharides of edible and medicinal resources. The manuscript, from my point of view, is publishable in its current form if some minor issues are attended:
- Line 184, 458, 473, 593, 598, and 601: In vivo and/or in vitro lacks italic font.
- Figures 1, 2, and 3: The font in figures is small; consider making it more prominent to better reading comprehension.
Author Response
Dear reviewer,
I am very grateful to your comments for the manuscript. According with your advice, we amended the relevant part in manuscript. We sincerely hope that the revision meets your satisfaction. Here below are our responses to your comments.
1) Line 184, 458, 473, 593, 598, and 601: In vivo and/or in vitro lacks italic font.
Response: We were really sorry for our mistakes and thank you for your reminder. We have changed all in vivo and in vitro in this article to italic font.
The revisions were highlighted with red.
2) Figures 1, 2, and 3: The font in figures is small; consider making it more prominent to better reading comprehension.
Response: We are very sorry for the unclear picture we provided. We've increased the font size and increased the resolution of the Figures.
The revisions were highlighted with red.
Reviewer 4 Report
Manuscript ID: biomolecules-2276514
The paper entitled “The Potential of Edible and Medicinal Resources Polysaccharides for Prevention and Treatment of Neurodegenerative Diseases”. In this review, the authors reviewed mechanisms and pathways of Edible and Medicinal Resources (EMPs) against neurodegenerative diseases. I have cheeked the literature, this is an interesting paper, and manuscript is well organized. However, for publication a minor revision is needed.
I have just few comments in order to improve the final version of the manuscript before publication, please find them below:
Line 157: C. elegans should be in italic.
Table 1: the table must be organized according to class of polysaccharides
Table 1: Row 2, 14 and 28 the plant names should be added.
Table 1: Row 47, the full name should be added.
Figures: Low resolution (should have a higher resolution)
Author Response
Dear reviewer,
I am very grateful to your comments for the manuscript. According with your advice, we amended the relevant part in manuscript. We sincerely hope that the revision meets your satisfaction. Here below are our responses to your comments.
1) Line 157: C. elegans should be in italic.
Response: We were really sorry for our mistakes and thank you for your reminder. We have changed all C. elegans in this article to italic font.
The revisions were highlighted with red.
2)Table 1: the table must be organized according to class of polysaccharides
Table 1: Row 2, 14 and 28 the plant names should be added.
Table 1: Row 47, the full name should be added.
Response: Thank you for your comments. Per your suggestion, we have reorganized the table according to the class of polysaccharides and added the complete Latin names of all the sources of polysaccharides.
The revisions were highlighted with red.
3)Figures: Low resolution (should have a higher resolution).
Response: We are very sorry for the unclear picture we provided. We've increased the resolution of the Figures.
The revisions were highlighted with red.
Round 2
Reviewer 2 Report
I accept the introduced corrections. I like the table added to the publication. It is clear and comprehensive. I would consider including it in the text of the manuscript but I leave that to the authors' decision.